# Anxiety and Depression and Related Risk Factors in Italian Healthcare Providers Involved in Adverse Events

**DOI:** 10.3390/healthcare13030343

**Published:** 2025-02-06

**Authors:** Isolde Martina Busch, Maria Angela Mazzi, Fiammetta Cosci, Loretta Berti, Veronica Marinelli, Francesca Moretti, Olga Maggioni, Albert W. Wu, Michela Rimondini

**Affiliations:** 1Department of Neurosciences, Biomedicine and Movement Sciences, Section of Clinical Psychology, University of Verona, 37134 Verona, Italy; mariangela.mazzi@univr.it (M.A.M.); loretta.berti@univr.it (L.B.); olga.maggioni@univr.it (O.M.); michela.rimondini@univr.it (M.R.); 2Department of Health Sciences, University of Florence, 50121 Florence, Italy; fiammetta.cosci@unifi.it; 3Department of Engineering for Medicine Innovation, University of Verona, 37134 Verona, Italy; veronica.marinelli@univr.it; 4Department of Neurosciences, Biomedicine and Movement Sciences, Section of Movement Sciences, University of Verona, 37134 Verona, Italy; francesca.moretti@univr.it; 5Department of Health Policy and Management, Johns Hopkins Bloomberg School of Public Health, Baltimore, MD 21205, USA; awu@jhu.edu

**Keywords:** occupational mental health, healthcare workers, adverse events, second victims, risk factors, screening, depression, anxiety

## Abstract

**Background/Objectives:** Despite the importance of the second victim phenomenon for healthcare systems, there is limited research on Italian healthcare providers. We assessed emotional distress in individuals impacted by an adverse event using the Withstand-PSY Questionnaire (WS-PSY-Q). Additionally, we aimed to identify potential risk factors for anxiety and depression. **Methods**: A cross-sectional online survey of 284 participants. Measures included the WS-PSY-Q, Beck Depression Inventory-II (BDI-II), and State-Trait Anxiety Inventory (STAI-Y). Descriptive analyses and seemingly unrelated regression, jointly estimating anxiety and depression, were conducted using Stata (version 18). **Results**: Fifty-nine percent of the participants tested positive for anxiety (WS-PSY-Q anxiety subscale ≥ 16), thirty-seven percent for depression (WS-PSY-Q depression subscale ≥ 22), and thirty-five percent for both. In the final model, anxiety symptoms following the adverse event were associated with pre-event anxiety levels (*p* < 0.01), seeking psychological help (*p* < 0.05), self-perceived responsibility (*p* < 0.01), severity of the adverse event for the patient (*p* < 0.05), and punitive workplace climate (*p* < 0.05). Correlates of post-event depressive symptoms included pre-existing depression (*p* < 0.01), self-perceived responsibility (*p* < 0.01), severity of the impact of the adverse event (*p* < 0.01), punitive or neutral workplace climate (*p* < 0.05), and seeking psychological help (*p* < 0.01). **Conclusions**: This study adds to the growing understanding of the mental health difficulties that healthcare workers in Italy encounter after adverse events, addressing both individual and systemic risk factors. Proactive implementation of mental health measures for healthcare workers could foster their well-being after adverse events and promote a stronger, more just organizational culture.

## 1. Introduction

Despite the mandate for every healthcare professional to “primum non nocere” (first, do no harm) [1], in healthcare practice there is always the risk of adverse events [2]. An adverse event is defined as an unexpected and unintentional incident that can cause harm to a patient, possibly leading to temporary or permanent disability [3].

An adverse event may be due to a medical error and is thus preventable (i.e., a mistake made by a healthcare provider or within a healthcare system, resulting from incorrect actions, such as the administration of a wrong medication, or the omission of necessary action, such as failing to provide adequate treatment) or due to factors which are not preventable (e.g., an adverse reaction to a correctly administered drug). Thus, a medical error may be a potential cause of an adverse event, but not all adverse events result from error [4,5]. Frequent types of medical errors include surgical errors, which carry the highest risk of severe patient injury and death, as well as misdiagnoses, medication errors, healthcare-associated infections, problems associated with medical devices, and ineffective communication [6].

When there is an adverse event, its repercussions can affect patients and their families, jeopardizing the physical and mental health of patients and placing stress on family dynamics and financial resources. In certain instances, adverse events have a fatal outcome [7,8,9,10,11]. In addition, adverse events can have serious effects on the healthcare providers involved, potentially harming their professional and personal quality of life, and can generally impose a major challenge to the healthcare system, manifesting as financial costs, decreased productivity, and a weakened trust in the healthcare system [3,12,13,14,15,16,17]. Recognizing these multifaceted consequences, Wu coined the term “second victim” [16] to describe healthcare providers who are not only involved in but also harmed by the occurrence of the adverse event. Psychological symptoms reported by second victims include troubling memories, remorse, guilt, shame, anxiety, and depression and physical symptoms, such as sleeping difficulties and a change in appetite [18]. It has been suggested that the psychological impact can be affected by various factors, such as individual characteristics, the time since the event, the severity of harm, and the safety culture within the healthcare worker’s organization [19]. Working in an organization characterized by a blame culture might inhibit disclosure of adverse events and suppress communication about adverse events among colleagues who may fear retribution or criticism [20,21,22]. Regarding communication within the team, if the adverse event exposes gaps in communication processes, team members may be hesitant to share critical information in the future, fearing it could lead to additional communication errors. Fostering a culture of learning from errors, encouraging open, transparent communication, providing managerial support, and adopting a distributed and transformational leadership style can lay the foundation for more effective management of adverse events [20,23,24]. This creates a more favorable environment for healthcare providers [20].

Errors and adverse events are still too often addressed by focusing on the individual involved (person approach) rather than on the system in which the event occurs (systemic approach) [25]. The person approach focuses on the individual who appears to have committed the error, e.g., blaming a healthcare provider for inattention or forgetfulness. The system approach considers the underlying systemic conditions, with the goal of preventing recurrence or mitigating its consequences [25]. Accordingly, the system approach encourages a “just culture” that focuses on system failures and does not unfairly blame individuals but still holds them accountable [25,26,27,28]. In this way, organizational culture can significantly impact the well-being and professional growth of healthcare providers [29].

While these ideas are becoming more accepted in medical practice, many countries are lagging in research into different aspects of the second victim phenomenon. For example, in Italy, there is a need for more thorough investigation, as only a few studies on the second victim phenomenon have been carried out and adequate support structures are still missing [30,31,32,33].

Closing these research gaps is crucial because, as outlined above, the issue has a significant impact on well-being and job performance of physicians, the entire working team, and, consequently, the overall quality of healthcare.

Our aim was to assess emotional distress in healthcare providers throughout Italy who were involved in an adverse event. Specifically, we wanted to understand their levels of anxiety and depression and explore potential risk factors at both the individual (e.g., self-perceived responsibility) and system level (e.g., workplace climate) associated with these symptoms.

## 2. Materials and Methods

This was a cross-sectional study on 284 healthcare providers in Italy who were involved in an adverse event. Quantitative data were collected between September 2021 and June 2022 using an anonymous online survey on the LimeSurvey platform. The study design includes the use of three questionnaires: two for general screening and a third one specifically focusing on occupational mental health. The analyses presented here are part of a larger study that aimed to validate the Withstand-PSY Questionnaire (WS-PSY-Q) using a clinimetric approach [34].

### 2.1. Ethical Considerations

Approval for the study was obtained from the Institutional Review Board at the University of Verona (nr. 03/2021). All procedures contributing to this work comply with the ethical standards of the relevant national and institutional committees on human experimentation and with the Helsinki Declaration of 1975, as revised in 2013. Participants were asked to provide written consent and were guaranteed anonymity.

### 2.2. Recruitment Procedure

In brief, we collaborated with the Italian National Institute of Health, Department of Social Neurosciences, to recruit healthcare providers who were involved in a harmful incident or a no harm incident in the past [35], while working in clinical healthcare settings (e.g., inpatient, outpatient) across Italy. We used purposive sampling (i.e., chain sampling). Specifically, the online survey was promoted on the websites of the respective healthcare institutions associated with the Italian National Institute of Health and through phone/social media contacts of the research group. Additionally, the link to the online survey was shared via email with healthcare staff through the internal mailing lists of the contacted hospitals and healthcare centers, accompanied by a short video outlining the study’s aims. Reminder emails were sent over a 3-month period.

### 2.3. Inclusion Criteria

Inclusion criteria were as follows: (1) working in a healthcare institution (e.g., hospital, nursing home, private practice); (2) having been involved in the past in a harmful incident (i.e., an event that resulted in harm to a patient) or no harm incident (i.e., an event that reached a patient, but no discernable harm resulted) [36,37]; (3) Italian speaker; (4) having signed the informed consent form.

### 2.4. Measures

The online survey included a short project description, the WS-PSY-Q [34], the Beck Depression Inventory-II (BDI-II) [38,39,40], and S-Anxiety Scale of the State-Trait Anxiety Inventory—Form Y (STAI-Y) [41,42].

The Withstand-PSY Questionnaire (WS-PSY-Q) is a cross-sectional questionnaire divided into three sections (General Information, Emotional Distress, Coping) (Appendix A). The current study focuses on the first two sections. The General Information section collects socio-demographic characteristics (e.g., gender, age, years of work experience) and information related to the adverse event and its consequences (e.g., time of occurrence, severity of the consequences, workplace climate) to enhance understanding and contextualization of the situation. As regards the severity of the consequences of the adverse event, the following categorization is applied [43]: Level 1, minor outcome (events that led to temporary harm requiring minor therapeutic interventions, such as a minimal sedation after incorrect drug dosage); Level 2, moderate outcome (events that led to temporary harm with the need for hospitalization or prolonged hospitalization, such as a post-surgical complication requiring prolonged hospitalization); Level 3, significant outcome (events that contributed to permanent disability, such as a surgical procedure error causing nerve damage and resulting in loss of sensation); Level 4, severe outcome (need for life-saving interventions, such as an anaphylactic shock following drug administration requiring intensive care admission); Level 5, death (events that contributed to the patient’s death, such as a diagnostic delay leading to delayed treatment and subsequent death). The Emotional Distress scale includes 24 items divided into anxiety and depression subscales to retrospectively assess healthcare providers’ psychological symptoms before and after the adverse event. This allows evaluation of both the current emotional state of healthcare providers and the emotional impact of the adverse event. Accordingly, there are two pairs of cut-off scores. The cut-off scores for the anxiety and depression subscales were found to be ≥16 and ≥22, respectively, regarding current emotional state. Regarding the emotional impact of the adverse event, the cut-off scores for the anxiety and depression subscales are ≥2 and ≥3, respectively.

Cronbach’s alpha was 0.91 (bootstrap 95%CI = 0.89–0.92) and 0.90 (bootstrap 95%CI = 0.88–0.92) for anxiety and depression subscales, respectively.

The Beck Depression Inventory-II (BDI-II) [38,39,40] and the S-Anxiety Scale of the State-Trait Anxiety Inventory—Form Y (STAI-Y) (STAI-Y) [41,42] were also administered. The BDI-II is a 21-item self-administered questionnaire. Key symptoms of depression are evaluated using a rating scale from 0 to 3. A higher total score suggests increased severity of depressive symptoms. Cronbach’s alpha was 0.93 (bootstrap 95%CI = 0.92–0.95).

The STAI-Y is a commonly used tool for assessing both trait and state anxiety. The anxiety levels are measured on a 4-point scale, where higher scores reflect higher levels of anxiety. Cronbach’s alpha was 0.95 (bootstrap 95%CI = 0.94–0.96).

### 2.5. Statistical Analyses

To assess participants’ emotional distress and to identify potential risk factors for anxiety and depression, we performed the following analyses, using Stata (version 18).

We calculated the prevalence rates of anxiety and depression by participant and adverse event characteristics, by using the established cut-offs, and 95% confidence intervals (CI) were applied.

We also calculated the comorbidity rates, reporting scores for the WS-PSY-Q subscales and STAI-Y and BDI-II. We conducted one-sided paired *t*-tests to compare the pre- and post-adverse event scores for each item (4-point Likert scale) of the WS-PSY-Q anxiety and depression subscales.

Multivariate preliminary explorations were conducted on the overall scores of the scales (i.e., anxiety and depression subscales of the WS-PSY-Q, anxiety scale of the STAI-Y, depression scale of the BDI-II) and not on the dichotomous categorization of emotional distress (i.e., screened positive or negative for anxiety/depression) to retain as much collected information as possible. Specifically, seemingly unrelated regression analysis [44] was utilized, in which a system of four linear regressions was resolved to assess the impacts of a set of influencing factors (i.e., psychological characteristics and socio-demographic characteristics of participants, characteristics of the AE). This approach is valuable for enhancing regression estimates when assuming that all predictors are independent variables, while accounting for the possibility that errors across different equations may be correlated. This is particularly relevant as the outcomes are derived from questionnaires measuring distinct yet related aspects of emotional distress (e.g., symptoms of anxiety and depression). Subsequently, the Breusch–Pagan test was carried out as a post hoc analysis to examine the dependency of regression residuals. A stepwise process was applied to select potential predictors, considering the three blocks of variables mentioned above. The variables that passed the filter of the preliminary analyses were included in the final model.

## 3. Results

Our sample consisted of 284 participants, all of whom reported being involved in an adverse event, of which 81% were women. Forty-six percent were nurses/midwives, and twenty-seven percent were physicians. The majority (52%) had more than 20 years of work experience. More than two-thirds (76%) of participants worked in hospitals, 18% in other healthcare settings (e.g., private practices, nursing homes), while the remaining 16% chose not to answer this question. Regarding geographical distribution, 71% of participants worked in northern Italy, 3% in central Italy, and 43% in southern Italy, while 10% of participants did not respond.

Table 1 and Table 2 show the prevalence rates of anxiety and depression based on participant characteristics and characteristics of the adverse event, respectively, for the anxiety and depression subscales of the Withstand-PSY Questionnaire (WS-PSY-Q), the State-Trait Anxiety Inventory—Form Y (STAI-Y), and the Beck-Depression Inventory-II (BDI-II). In the entire sample, 59% of participants tested positive for anxiety on the WS-PSY-Q subscale (cut-off of ≥16) and 58% on the STAI-Y (cut-off of ≥40). As regards depression, the prevalence for both the WS-PSY-Q subscale (cut-off of ≥22) and the BDI (cut-off of ≥10) was 37%.

A total of 35% of the participants exhibited both anxiety and depression. While 24% of second victims experienced only anxiety, 2% of second victims had only depression (see Appendix A).

Figure 1 shows the pre- and post-adverse event scores for the two WS-PSY-Q subscales across four groups: participants who scored negative for anxiety or depression and those who tested positive. These scores represent participants’ perceptions of emotional distress before (i.e., pre-adverse event score) and after (i.e., post-adverse event score) the adverse event. The figure shows that, not only did the two groups testing positive show an increase in symptoms after the adverse event, but the two groups testing negative did also, although to a lesser extent.

Approximately 29% of healthcare providers displayed clinically significant symptoms of anxiety before the adverse event, and this percentage increased to 59% post-event. Regarding depression, 20% of participants had clinically significant symptoms before the event, while 37% reported them afterwards (see Appendix A).

The differences between the pre- and post-scores of the individual items were significant for all items (*p* < 0.01) (see Appendix A). Regarding the WS-PSY-Q anxiety subscale, items 2 (“I feel anxious and concerned.”), 3 (“I feel worried.”), and 5 (“I feel embarrassed.”) exhibited the largest mean differences of 0.48, 0.41, and 0.32 points, respectively. The items with the greatest mean differences on the WS-PSY-Q depression subscale were items 10 (“I feel angry.”), 4 (“I feel regret and remorse.”), and 6 (“I feel guilty.”), with differences of 0.46, 0.43, and 0.40 points, respectively.

As shown in Table 3, which presents the set of preliminary seemingly unrelated regression models, the psychological variables explained about 60% of the variation of the WS-PSY-Q subscales and approximately 35% of the variation of the general screening instruments. Participants’ socio-demographic characteristics accounted for less than 10% of the variation in anxiety and depression. The characteristics of the adverse event have a stronger predictive capacity for the WS-PSY-Q subscales (0.17 and 0.23 for anxiety and depression, respectively) compared to STAI-Y and BDI-II, which only have an R^2^ of 12%. (see Appendix A).

The final seemingly unrelated regression model (see Figure 2) depicts the four outcome variables as external squares: depression (measured by the WS-PSY-Q depression subscale and the BDI-II) and anxiety (measured by the WS-PSY-Q anxiety subscale and the STAI-Y). These outcome variables are predicted by the risk factors displayed at the center of the figure.

The model indicates that symptoms of anxiety following the adverse event, measured by the WS-PSY-Q anxiety subscale, were significantly associated with pre-event anxiety, seeking psychological help, self-perceived responsibility, severity of the consequences of the adverse event, and a punitive workplace climate. Symptoms of anxiety, as measured by the STAI-Y, were significantly associated with pre-event anxiety (measured with the WS-PSY-Q anxiety subscale), the request for psychological help, self-perceived responsibility, and a punitive workplace climate.

Risk factors for post-event depressive symptoms, as measured by the WS-PSY-Q depression subscale, included pre-event depression, seeking psychological help, self-perceived responsibility, the severity of the consequences of the event, and a punitive or neutral workplace climate. Symptoms of depression, as measured by the BDI-II, could be predicted by pre-event depression, seeking psychological help, 5–10 years of work experience, and a punitive workplace climate (see Appendix A).

The post hoc analysis of the regressions revealed that the correlations among the residuals ranged from 0.32 to 0.67. The Breusch–Pagan test of independence confirmed the rejection of independence (chi^2^(6) = 457.3, *p* < 0.01), thus supporting the use of seemingly unrelated regression models to obtain more efficient estimates.

## 4. Discussion

This paper adds to the increasing knowledge base on the second victim phenomenon and is one of the first studies to investigate it in Italy. Specifically, we aimed to assess healthcare providers’ levels of anxiety and depression following adverse events and explore potential individual and system risk factors for the development of emotional distress.

The results revealed significant levels of anxiety and depression among the affected employees and identified various risk factors for poor mental health. There appear to be specific risk factors and areas for improvement at both the individual (e.g., healthcare provider well-being before events) and system levels (e.g., work climate). Healthcare institutions should carefully consider these factors in their management to ensure high standards of quality and safety in the care process.

At the system level, we identified a punitive workplace environment as a risk factor for anxiety and depression. The Withstand-PSY Questionnaire (WS-PSY-Q) revealed that even a neutral atmosphere is linked to depressive symptoms. These results emphasize the need to move away from a culture of blame, where individuals are targeted and blamed for negative events that are frequently the result of systemic errors rather than individual faults [21,25,45]. The findings highlight the importance of a supportive workplace environment in buffering emotional distress among healthcare providers. A culture that remains neutral and neither assigns blame nor takes responsibility for the systemic factors contributing to adverse events, and does not legitimize or support the suffering of second victims, can also be detrimental to professional well-being. A neutral culture that does not punish but remains indifferent to providers’ suffering is not enough. We urge Italian healthcare institutions to work towards the promotion and implementation of a supportive and just culture that improves the handling of critical clinical situations, including adverse events, medical errors, and workplace violence, while promoting respect in the workplace [28,46,47,48,49].

Reducing blame culture is necessary but insufficient to achieve the right balance in both supporting second victims and acknowledging their individual responsibilities in adverse events. A just culture should prevent healthcare providers from making mistakes, support them afterward, and encourage a sense of responsibility where individuals are aware of the consequences of their actions in the process of delivering care.

Our analyses indicate that feeling responsible for the adverse event, which could mirror actual responsibility and associated self-blame, is a risk factor for both depression and anxiety [50]. Acknowledging real responsibility can be more painful than avoiding it but it is a necessary discomfort. It is distinct from blaming oneself unfairly, which can trigger defensiveness or avoidance [51]. Instead, feeling regret and owning up to one’s responsibility can inspire us to grow and aim for better results.

This perception of self-blame is influenced by the blame culture that is still prevalent in many institutions, but it may extend further than that. In healthcare, time and space are often limited for processing experiences and emotions and healthcare providers may not be used to identifying and dealing with their own feelings. This oversight can cause confusion between actual and perceived responsibilities, leading to stress and a sense of overwhelming personal responsibility for the adverse event [52]. Addressing these emotions in psychological support programs is expected to result in improved emotional well-being, a greater sense of control over one’s inner state [52], and increased self-awareness of personal responsibilities in the events that occurred. Psychological interventions for second victims could also positively affect other workplace issues that might play a particular role in the aftermath of an adverse event, such as absenteeism (i.e., missing work due to various reasons), presenteeism (i.e., when individuals show up to work despite being in poor physical or mental health [53]), and resenteeism (i.e., staying in a job with low enthusiasm [54]). In some cases, all these conditions can be seen as individual strategies adopted to cope with the feeling of being left alone to deal with a destabilizing professional challenge.

The significance of addressing the psychological well-being of healthcare providers is supported by the current findings, emphasizing the importance of taking action both before and after an incident occurs. Elevated anxious and depressive symptoms before the adverse event predicted anxiety and depression after it, showing that individual vulnerabilities play a significant role. It is essential for healthcare leaders and managers to prioritize mental health and resilience of healthcare workers and to implement preventive measures [55]. Periodic screening of healthcare providers for emotional distress should be prioritized in the healthcare sector. Additionally, having mental health staff in various units of hospitals could enhance the early detection of psychological distress among the workforce. These programs can assist in interventions to aid healthcare providers in developing effective strategies for processing emotions. This can enhance their well-being, potentially reducing the risk of burnout and emotional distress [56,57,58]. The widespread implementation of such programs would also help reduce stigma and make feelings more accepted in the healthcare workforce [59,60]. This would be highly important because healthcare providers are often expected to remain neutral and objective when dealing with mortality and morbidity and any display of emotions may be perceived as a weakness, indicating a potential lack of objectivity and resilience [52]. Finally, reducing personal discomfort helps healthcare providers to approach the risk of making errors calmly, rather than with fear or anxiety. It also helps them to be prepared to accept the consequences of their mistakes without feeling overwhelmed, avoiding the tendency to blame others or fall into depression.

When healthcare providers prioritize their well-being and that of their colleagues, they demonstrate commitment to their professional duties and enhance public trust in the healthcare system [61]. This, in turn, can improve communication between healthcare providers and patients, as well as their families, and promote patient-centered care [62,63,64].

Participants who mentioned considering seeking psychological help displayed emotional distress in the WS-PSY-Q and the two general screening instruments. Among these, only the WS-PSY-Q revealed a connection between those who actively sought psychological help and higher levels of distress. Similarly, the severity of the consequences of the adverse event was only identified as a risk factor for emotional distress measured by the WS-PSY-Q and did not show any link to levels of emotional distress measured by the two general screening instruments. These findings suggest that only the WS-PSY-Q, which focuses on the second victim phenomenon, accurately captures the different factors related to healthcare providers’ mental health. The only exception to this trend was healthcare providers’ work experience of 5 to 10 years which predicted only depressive symptoms measured by the Beck Depression Inventory-II (BDI-II). Compared to healthcare providers with only a few years of work experience, those with 5–10 years of work experience might begin to feel the cumulative pressure of the highly demanding medical environment. However, they may have fewer coping strategies compared to their more experienced colleagues, making them particularly susceptible to emotional exhaustion and depressive symptoms, a more general condition which in this case might have been captured best by the BDI-II. However, further investigation is needed to understand the exact reasons behind this result.

### 4.1. Limitations

It is important to note that the responses from our sample may be affected by different biases (e.g., recall bias, self-selection bias, nonresponse bias, social desirability bias) [34]. Specifically, by retrospectively evaluating healthcare providers’ emotional distress before and after the adverse event, the questionnaire is at risk of recall bias. Moreover, by using chain sampling, we cannot accurately report the rate of second victims in Italian healthcare institutions. However, this recruitment technique, which effectively connects with hard-to-reach groups, proved beneficial in identifying our target population. Furthermore, self-selection and nonresponse bias could have led to an inaccurate portrayal of the frequency of psychological symptoms [31]. Healthcare providers significantly impacted by the adverse event might have been more inclined to participate in the survey, potentially skewing the results. Additionally, the reasons for nonparticipation are unknown. It is possible that nonresponders were not involved in an adverse event, were not psychologically affected by one, or used avoidance strategies to cope with their second victim experience. Social desirability bias could have also affected the responses [34]; for instance, some participants might have downplayed their symptoms due to feeling the need to appear strong in their role as healthcare providers.

### 4.2. Future Research Directions

In the future, we aim to analyze how participants coped with the event and explore whether different types of coping could mediate the risk for emotional distress in second victims. It would be also interesting to investigate the link between emotional distress in healthcare providers and other mental health conditions they may face, such as burnout, compassion fatigue, and moral injury [65,66]. Large, prospective studies could provide clearer evidence on the frequency and intensity of the second victim phenomenon, as well as its long-term impact on the mental health of those affected. Where psychological support structures are available, whether as preventive interventions or as assistance following adverse events, it would be important to assess their direct and indirect benefits for second victims, patients and caregivers, and healthcare centers [55,59]. Specifically, for second victims, such interventions should be evaluated not only in terms of emotional outcomes but also through clinical and organizational outcomes, such as reduced number of adverse events, absenteeism, and workforce turnover.

## 5. Conclusions

The present study adds to the growing understanding of the mental health difficulties experienced by healthcare workers after adverse events. We found several factors that can contribute to emotional distress among those affected, both on an individual and systemic level. The results indicated the importance of introducing proactive mental health initiatives for healthcare workers, monitoring their well-being following adverse events, and cultivating a strong culture of safety centered on promoting responsibility rather than guilt.

## Figures and Tables

**Figure 1 healthcare-13-00343-f001:**
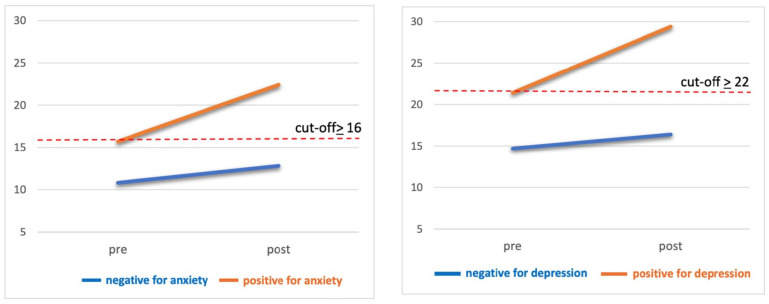
Pre- and post-adverse event scores for anxiety and depression subscales of the Withstand-PSY Questionnaire (WS-PSY-Q) in four groups (i.e., participants who screened negative for anxiety or depression, and participants who screened positive for anxiety or depression). Notes. WS-PSY-Q anxiety subscale cut-off of ≥16 for current emotional state; WS-PSY-Q depression subscale cut-off of ≥22 for current emotional state.

**Figure 2 healthcare-13-00343-f002:**
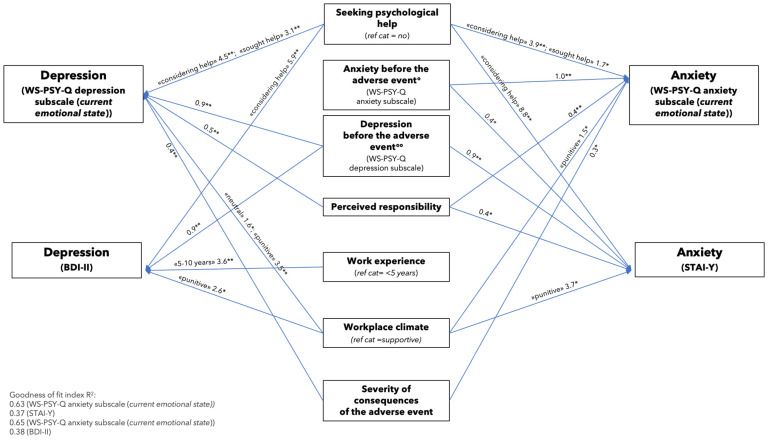
Seemingly unrelated regressions for anxiety and depression: path diagram. Abbreviations. BDI-II = Beck Depression Inventory-II; ref cat = reference category; STAI-Y = State-Trait Anxiety Inventory—Form Y; WS-PSY-Q = Withstand-PSY Questionnaire. Notes. * *p* ≤ 0.05; ** *p* ≤ 0.01. ° Measured with the WS-PSY-Q anxiety subscale referring to the emotional state before the adverse event; °° measured with the WS-PSY-Q depression subscale referring to the emotional state before the adverse event; the arrows in the final model represent the direction of the relationships between risk factors and outcome variables.

**Table 1 healthcare-13-00343-t001:** Prevalence of anxiety and depression by participant characteristics (with 95% confidence interval).

	Anxiety		Depression	
	WS-PSY-Q Anxiety Subscale ° (Current Emotional State)	STAI-Y *	WS-PSY-Q Depression Subscale °° (Current Emotional State)	BDI-II **
**Whole sample**	59% (52.8–64.6)	58% (52.5–64.2)	37% (31.3–42.9)	37% (31.7–43.22)
**Gender**				
Male	42% (28.7–55.9)	45% (32.0–59.5)	24% (13.2–37.0)	24% (13.2–37.0)
Female	63% (56.3–69.2)	62% (54.9–67.9)	40% (33.8–46.8)	41% (34.2–47.3)
**Age**				
18–30	81% (61.9–93.7)	70% (49.8–86.3)	44% (25.5–64.7)	48% (28.7–68.1)
31–40	60% (46.0–73.6)	68% (53.7–80.1)	40% (26.5–54.0)	42% (28.1–55.9)
41–50	55% (45.2–63.6)	54% (44.4–62.8)	35% (26.3–43.1)	36% (27.1–44.8)
51–60	55% (43.4–67.3)	56% (43.4–67.3)	36% (25.1–48.3)	35% (23.9–46.9)
>60	64% (30.8–89.1)	55% (23.4–83.3)	36% (10.9–69.2)	27% (6.0–61.0)
**Profession**				
Nurse/midwife	60% (51.0–68.3)	57% (47.9–65.4)	38% (29.6–46.7)	39% (38.3–47.5)
Physician	68% (46.4–78.1)	62% (49.8–72.3)	44% (32.4–55.3)	41% (30.0–52.8)
Psychologist/psychotherapist	44% (21.5–69.2)	56% (30.8–78.5)	39% (17.3–64.3)	22% (6.4–47.6)
Healthcare assistant	44% (21.5–69.2)	33% (13.3–59.0)	6% (0.1–27.3)	17% (3.6–41.4)
Rehabilitation support worker	57% (34.0–78.2)	67% (43.0–85.4)	38% (18.1–61.6)	43% (21.8–66.0)
Other	41% (18.4–67.1)	76% (50.1–93.2)	29% (10.3–60.0)	41%2 (18.4–67.1)
**Work experience, years**				
<5	71% (53.7–85.4)	63% (44.9–78.5)	37% (21.5–55.1)	34% (19.1–52.2)
5–10	63% (43.9–80.1)	63% (43.9–80.1)	43% (25.5–62.6)	50% (31.3–68.7)
10–20	56% (43.3–67.6)	59% (46.2–70.2)	36% (24.6–48.1)	39% (27.2–51.0)
>20	56% (48.0–64.5)	56% (48.0–64.5)	36% (28.5–44.5)	35% (27.3–43.1)
**Self-perceived responsibility**				
Low (0–3)	44% (36.4–51.9)	52% (44.6–60.1)	25% (18.7–32.3)	32% (24.6–39.2)
Medium (4–6)	73% (61.4–82.7)	62% (50.1–73.2)	50% (38.1–61.9)	41% (30.5–53.9)
High (7–10)	93% (80.5–98.5)	76% (60.6–88.0)	62% (45.6–76.4)	52% (36.4–68.0)
**Seeking psychological help**				
No	56% (49.5–62.9)	53% (46.4–59.9)	32% (25.5–38.1)	33% (27.2–40.0)
No, but I am thinking about it	79% (57.9–92.9)	88% (67.6–97.3)	58% (36.6–77.9)	71% (48.9–87.4)
Yes	60% (43.4–76.0)	71% (54.1–84.6)	55% (38.3–71.4)	39% (24.0–56.6)

Abbreviation. BDI-II = Beck Depression Inventory-II; STAI-Y = State-Trait Anxiety Inventory—Form Y; WS-PSY-Q = Withstand-PSY Questionnaire. Notes. * STAI-Y cut-off of ≥40; ** BDI-II cut-off of ≥10; ° WS-PSY-Q anxiety subscale cut-off of ≥16 for current emotional state; °° WS-PSY-Q depression subscale cut-off of ≥22 for current emotional state.

**Table 2 healthcare-13-00343-t002:** Prevalence of anxiety and depression by characteristics of the adverse event (with 95% confidence interval).

	Anxiety		Depression	
	WS-PSY-Q Anxiety Subscale ° (Current Emotional State)	STAI-Y *	WS-PSY-Q Depression Subscale °° (Current Emotional State)	BDI-II **
**Time of occurrence**				
<1 month ago	74% (56.7–87.5)	83% (66.4–93.4)	60% (42.1–76.1)	57% (39.4–73.7)
2–12 months ago	64% (53.3–73.5)	63% (52.2–72.5)	41% (31.4–52.1)	41% (31.4–52.1)
>1 year ago	52% (44.1–60.3)	50% (42.2–58.4)	29% (22.0–36.9)	30% (23.2–58.2)
**Adverse event related to COVID-19 pandemic**				
No	58% (49.6–66.6)	55% (46.7–63.8)	35% (26.7–43.1)	36% (28.1–44.5)
Management of a COVID-19 patient	54% (42.1–65.5)	59% (47.3–70.4)	36% (24.9–47.3)	45% (33.3–56.6)
Implementation of infection control and containment procedure	62% (46.5–76.2)	60% (44.3–74.3)	38% (23.8–53.5)	31% (18.2–46.7)
Other	71% (48.9–87.4)	71% (48.9–87.4)	54% (32.8–74.5)	33% (15.6–55.3)
**Severity of the consequences**				
Event without harm	49% (39.3–58.9)	50% (40.2–59.8)	23% (15.6–32.3)	31% (22.1–40.2)
Level 1 *, minor outcome	59% (45.7–71.5)	67% (54.0–78.7)	34% (22.7–47.7)	43% (30.0–55.9)
Level 2, moderate outcome	68% (51.4–82.5)	71% (54.1–84.6)	58% (40.8–73.7)	45% (28.6–61.7)
Level 3, significant outcome	50% (21.1–78.9)	42% (15.2–72.3)	42% (15.2–72.3)	42% (15.2–72.3)
Level 4, severe outcome	58% (27.7–84.8)	58% (27.7–84.8)	42% (15.2–72.3)	17% (2.1–48.4)
Level 5, death	74% (59.7–84.7)	60% (46.0–73.6)	51% (36.8–64.9)	43% (29.8–57.7)
**Workplace climate**				
Supportive	48% (37.6–58.4)	49% (38.6–59.4)	23% (15.0–32.6)	27% (18.5–37.1)
Neutral	59% (49.7–68.4)	56% (46.1–65.1)	33% (24.2–42.2)	35% (25.8–44.0)
Punitive	72% (60.4–81.8)	75% (63.3–84.0)	33% (24.2–42.2)	55% (42.8–66.2)

Abbreviation. BDI-II = Beck Depression Inventory-II; STAI-Y = State-Trait Anxiety Inventory—Form Y; WS-PSY-Q = Withstand-PSY Questionnaire. Notes. * STAI-Y cut-off of ≥40; ** BDI-II cut-off of ≥10; ° WS-PSY-Q anxiety subscale cut-off of ≥16 for current emotional state; °° WS-PSY-Q depression subscale cut-off of ≥22 for current emotional state.

**Table 3 healthcare-13-00343-t003:** Set of preliminary linear regressions of anxiety and depression scales, estimated by seemingly unrelated regression model: goodness of fit indices.

	WS-PSY-Q Anxiety Subscale (Current Emotional State)	STAI-Y	WS-PSY-Q Depresion Subscale (Current Emotional State)	BDI-II
**Psychological characteristics of participants**				
Anxiety before the adverse event °	✓	✓		✓
Depression before the adverse event °°		✓	✓	✓
Psychological help	✓	✓	✓	✓
Self-perceived responsibility	✓	✓	✓	
R^2^	0.62	0.36	0.62	0.35
RMSE	4.10	10.66	4.81	7.53
F (*p*-value)	179.9 **	33.4 **	167.3 **	53.2 **
**Socio-demographic characteristics of participants**				
Gender	✓			
Profession	✓		✓	
Work experience	✓			✓
R^2^	0.08	0.00	0.04	0.01
RMSE	6.43	13.17	7.65	9.30
F (*p*-value)	4.3 **	3173.3 **	4.0 **	3.7 *
**Characteristics of the adverse event**				
Distance from adverse event	✓	✓	✓	✓
COVID-19-related adverse event				
Severity of the consequences	✓	✓	✓	✓
Workplace climate	✓	✓	✓	✓
R^2^	0.17	0.12	0.23	0.12
RMSE	6.08	12.49	6.84	8.81
F (*p*-value)	11.4 **	7.3 **	16.9 **	7.3 **

Abbreviation. BDI-II = Beck Depression Inventory-II; STAI-Y = State-Trait Anxiety Inventory—Form Y; WS-PSY-Q = Withstand-PSY Questionnaire; Notes. The symbol ✓ indicates a significant regression coefficient (*p* ≤ 0.10); * *p* ≤ 0.05; ** *p* ≤ 0.01; ° measured with the WS-PSY-Q anxiety subscale referring to the emotional state prior to the adverse event; °° measured with the WS-PSY-Q depression subscale referring to the emotional state prior to the adverse event.

## Data Availability

The data presented in this study are available on reasonable request from the corresponding author.

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
