# Peer review of "Anxiety and Depression and Related Risk Factors in Italian Healthcare Providers Involved in Adverse Events"

_healthcare, 2025, doi:10.3390/healthcare13030343_

Round 1

Reviewer 1 Report

Comments and Suggestions for Authors

Thank you very much for the opportunity to review the manuscript ‘Analyzing anxiety and depression in healthcare providers involved in adverse events and related risk factors by using seemingly unrelated regression models.’

This study This study examines emotional distress among Italian healthcare providers affected by adverse events (AEs) using the WITHSTAND-PSY Questionnaire (WS-PSY-Q). It also explores potential risk factors for anxiety and depression. Based on a cross-sectional online survey of 284 participants, 59% showed anxiety symptoms, 37% depression symptoms, and 35% both. The study highlights the mental health challenges healthcare workers face after AEs and emphasizes the need for proactive mental health support and fostering a just workplace culture to enhance their well-being.

Overall, the paper is very well-written, scientifically sound and contains scientific novelty content. Also, the long duration of recruitment and correspondingly the number of examined patients are particularly noteworthy. Below, please find the comments with aspects that were found be in need of revision.

Title and abstract

The title of the study is apt and concise, it makes the reader eager to read the paper. However, the authors could consider adding “Italian” and leaving the last part to increase readership interest (“Analyzing anxiety and depression in Italian healthcare providers involved in adverse events and related risk factors”)

Introduction

The introduction is well-written and appropriately summarizes the introduction to the topic.

However, there are some indications that another revision by a native speaker should be conducted.

Also, it would be beneficial to provide an additional illustration of an AE - does this mean a medical error, for example, or what kind of errors are often mentioned and object of research.

In certain sections, the introduction could benefit from additional literature, such as on the psychosocial consequences of AEs (line 56) and on communication within the organization. Additionally, it would be worthwhile to explore the extent to which diverse leadership styles influence the psychological ramifications of AEs.

Materials and Methods

The description of the materials and methods is clear and easy to read.

However, a better introduction to the study would be provided if the study procedure were described first (i.e. that it was an online survey, when it was conducted, etc.). Even though this is a cross-reference, a brief explanation of the process and the sampling strategy would be helpful.

It is also not entirely clear to the reader which population is involved (i.e. what made them “second victims?”), from which hospitals in Italy or institutions the participants were drawn and how inclusion and exclusion criteria were chosen for participation. The authors could specify this in more detail. Also, the authors refer to “AE characteristics” (line 14), but it is not specified beforehand what exactly these are. It would be advisable to specify what is meant and what the authors have assessed. In addition, either in the text or in the supplementary materials, Cronbach's alpha of the questionnaires that have already been validated (BDI-II and STAI-Y) should be added.

Furthermore, the section from lines 123-134 should be backed up with more literature to support the choice of statistical methods.

Results

Unfortunately, it is not clear from the description of the results which basic population the sample is comprised of, e.g. whether it is from one or more hospitals. This should be added as described above.

Regarding Table 1 and the initial results, it is not initially clear to the reader to what extent the measured values are associated with the occurrence of AEs – was the influence on the participants recorded or distinguished in some way here?

Furthermore, it is often the case in manuscripts that abbreviations are not explicitly listed again at the beginning of each new section (e.g. when explaining the questionnaires). This should be checked and added throughout the manuscript.

Also, there is an extra bracket on line 166.

In Table 2, various degrees of severity of the consequences are described (1-5). It would be helpful here to explain, for example in the introduction, what these stages represent (is this a fixed classification?), as this would provide more insight into the internal logic of the AEs.

With regard to Figure 1 and the explanations that follow, it is not clear to the reader which pre-post analyses are being carried out here. The authors should describe this in more detail.

In figure 2, brackets are still missing in some places, a line is missing at the bottom right to close the box. The figure should therefore be revised by the authors, although it is generally very good and provides a good overview of the results.

Discussion

The discussion adequately addresses the research findings, the results are also supported by relevant literature.

Initially, the aim and purpose of the study should be re-stated in the discussion section.

The discussion describes some aspects that are examined in WS-PSY-Q. These are very interesting and relevant, but cannot be fully understood without prior knowledge. The contents of WS-PSY-Q should therefore be explained either in the introduction or at the relevant points in the discussion.

Also, the content described here in line 325 would also be clearer, as previously described, if it was described elsewhere which pre-post measurement it is.

In general, the discussion should be proofread again for spelling and language.

Reviewer 2 Report

Comments and Suggestions for Authors

Comments

English: Revise minor grammatical inconsistencies.

Introduction: Add and provide more specific examples or references to highlight the limited research in Italy (lines 74–76).

Methods: Elaborate on the participant recruitment process, especially regarding inclusion and exclusion criteria.

Include more details about the stepwise selection process for predictors to enhance clarity and reproducibility.

Results: Improve figure 1 and figure 2 as they are too small and have low resolution.

Elaborate on the implications of the Breusch-Pagan test results and their relevance to the SUR model.

Discussion: Elaborate on the rationale behind some findings (e.g., why 5–10 years of work experience predicts depressive symptoms).

Reviewer 3 Report

Comments and Suggestions for Authors thank you very much. it seems interesting manuscript.   I have listed my recommendations:   1-)be sure that seemingly unrelated is appropriate in the title of the study. you may consider removing it.   2-)you may choose not to abbreviate adverse events.   3-)you can mention more relevant literature in the introduction section.   4-)figure 2 seems unclear. the small texts are hard to read   5-)you can further discuss novel findings and mention contradicting factors in the discussion.   6-)you can add more suggestions for further studies.   7-)you can make following sentence specific to your study.   8-)some abbreviations in the manuscript may make it hard to understand.   9-)you should explain those biases as mentioned in the manuscript. It is important to note that the responses from our sample may be affected by different biases [16]   10-)its unclear why the reference is necessary. However, this recruitment technique, which has the advantage of reaching hidden and hard-to-reach populations, was useful in identifying our target population [41].   11-)you can add more recent references from the last 5 years.

Reviewer 4 Report

Comments and Suggestions for Authors

Introduction is much too short and lacks any explicit theoretical background. Much more coverage should be given to the relationship between AEs and adverse outcomes and to previous research supporting similar adverse outcomes for primary and secondary victims. The current Introduction is scarcely longer than the Abstract, and that is problematic. It shows a lack of engagement with the prior literature. What are the research hypotheses?

Method section needs to be improved in terms of detail and organization. More description of the participants (in its own subsection) as well as more description of each of the measures is necessary. Each measure should be described in enough detail for the reader to gain a thorough understanding of how the variable was operationally defined without referring the appendices.

Results section needs to be improved in terms of detail and organization. The reader is left to infer too much from the tables and figures without sufficient context in the main body. Are the regressions linear or logistic? It seems that the outcomes have been made dichotomous, so I assume the SURs are logistic regressions. It is not entirely clear, however. Either way, it would be good to report a greater range of statistical values associated with the analyses. What is the overall proportion of deviance in depression and anxiety diagnoses predicted by the full models? What is the direction of the relationship between each predictor and the outcome? What are the p-values associated with the overall model and individual predictors?

Round 2

Reviewer 3 Report

Comments and Suggestions for Authors

I accept it